# Flanking sequences influence the activity of TET1 and TET2 methylcytosine dioxygenases and affect genomic 5hmC patterns

Sabrina Adam[1], Julia Bräcker [2], Viviane Klingel[3], Bernd Osteresch[2], Nicole E. Radde[3], Jens Brockmeyer[2], Pavel Bashtrykov[1] & Albert Jeltsch [1✉]

TET dioxygenases convert 5-methylcytosine (5mC) preferentially in a CpG context into 5-hydroxymethylcytosine (5hmC) and higher oxidized forms, thereby initiating DNA demethylation, but details regarding the effects of the DNA sequences flanking the target 5mC site on TET activity are unknown. We investigated oxidation of libraries of DNA substrates containing one 5mC or 5hmC residue in randomized sequence context using single molecule readout of oxidation activity and sequence and show pronounced 20 and 70-fold flanking sequence effects on the catalytic activities of TET1 and TET2, respectively. Flanking sequence preferences were similar for TET1 and TET2 and also for 5mC and 5hmC substrates. Enhanced flanking sequence preferences were observed at non-CpG sites together with profound effects of flanking sequences on the specificity of TET2. TET flanking sequence preferences are reflected in genome-wide and local patterns of 5hmC and DNA demethylation in human and mouse cells indicating that they influence genomic DNA modification patterns in combination with locus specific targeting of TET enzymes.

[1] Institute of Biochemistry and Technical Biochemistry, Department of Biochemistry, University of Stuttgart, Allmandring 31, 70569 Stuttgart, Germany. [2] Institute of Biochemistry and Technical Biochemistry, Department of Food Chemistry, University of Stuttgart, Allmandring 5b, 70569 Stuttgart, Germany. [3] Institute for Systems Theory and Automatic Control, University of Stuttgart, Pfaffenwaldring 9, 70569 Stuttgart, Germany. ✉email: albert.jeltsch@ibtb.uni-stuttgart.de

DNA methylation at the C5-atom of cytosine residues (5mC) plays an essential role in gene regulation and chromatin biology[1,2]. DNA methylation levels are dynamically regulated by local DNA methylation and demethylation activity[3]. In mammals, active DNA demethylation is initiated by a TET dioxygenase-mediated oxidation of 5mC to 5-hydroxymethylcytosine (5hmC)[4–6] and the higher oxidation states 5-formylcytosine (5fC) and 5-carboxylcytosine (5caC)[7,8]. This is followed by the removal of 5fC and 5caC catalysed by thymine DNA glycosylase (TDG) and base excision repair[7]. Genomic 5hmC levels are in the range of 100–1000 ppm with the highest levels observed in ES cells and neurons, while genomic levels of 5fC and 5caC are 100–1000-fold lower[9]. Whole-genome analyses revealed that 5hmC on promoters, gene bodies, and transcription termination regions is positively correlated with gene expression, suggesting that 5hmC is a marker of active genes and it can play a role in stimulating gene expression by triggering DNA demethylation[10]. However, 5hmC also functions as a stable DNA modification beyond its role in DNA demethylation[11], and reading proteins for 5hmC have been identified as well[12,13], including the SRA domain of UHRF2[14,15]. Mammals contain three active TET paralogs[6], TET3 has a role in early development[16], while TET1 and TET2 are required at later stages during the development of primordial germ cells, somatic cell reprogramming and in the neural system[17]. Moreover, TET2 is often mutated in cancer[18] and a corresponding loss of 5hmC was reported for various cancer types[19].

TET enzymes are iron(II) dependent 5-methylcytosine dioxygenases using α-ketoglutarate as co-substrate. All three human TET family members share a highly conserved C-terminal catalytic domain with a double-stranded β-helix (DSBH) fold characteristic for this family of dioxygenases[17]. Crystal structures of TET2 with different bound DNA substrates[20–22] revealed that the methylated DNA binds above the DSBH core domain and is contacted by two protein loops. The target 5mC is flipped out of the DNA duplex and inserted into the active site cavity. The space emptied by the flipped base is filled by a loop containing a highly conserved tyrosine residue (Tyr1295 in human TET2), which stabilizes the G:C base pair of the hmCpG dinucleotide with H-bond and base-stacking interactions[20,22]. Consequently, TET2 has been shown to have the highest activity at mCpG sites, with mCpC[20] or mCpA[23] as the next preferred sequence context. Within the catalytic pocket of TET2, 5mC is specifically contacted by several interactions, which are necessary to hold the base in the extrahelical conformation required for catalysis[20]. The catalytic cavity is large enough to accommodate not only 5mC but also its derivatives 5hmC and 5fC, thus allowing for further oxidation steps in a sequential manner. Enzymatic assays revealed that the conversion of 5mC to 5hmC is 5–10-fold faster than the subsequent oxidation reactions of 5hmC to 5fC and 5fC to 5caC[8,21,22]. Structural data indicate that this difference is caused by a more constrained binding mode of the oxidized bases 5hmC and 5fC making the hydrogen abstraction step of the reaction less efficient[22]. The processivity of the stepwise oxidation process is currently unclear because processive[24] and distributive[25] reaction mechanisms have been reported.

Structural studies with TET2 revealed that the mCpG dinucleotide and DNA phosphate groups are involved in direct contacts with the enzyme. Initial biochemical studies detected no sequence selectivity for the DNA sequence besides their preference for the CpG dinucleotide[20]. However, novel experimental approaches recently revealed an unexpected level of flanking sequence effects on the activity, CpG recognition, and specificity of different mammalian DNA methyltransferases that is affecting cellular DNA methylation patterns[26–30]. Hence, further studies are needed to describe the potential effect of flanking sequences

on the activity of TET enzymes as well, in particular as DNMTs and TETs employ a related base flipping mechanism that includes large conformational changes of the DNA[31]. In this study, we used a Deep Enzymology approach[30] to investigate the flanking sequence effects on TET1 and TET2 using libraries of 5mC and 5hmC containing CpG and non-CpG substrates embedded in a randomized sequence context. Single-molecule readout of the oxidation levels of product molecules together with their specific sequence by bisulfite conversion coupled to NGS allows a very detailed analysis of the effects of all bases at all positions of the substrate on the catalytic activity of the TET enzymes. Our data show a pronounced effect of the flanking sequences from −3 to +2 position on the activity of TET1 and TET2, with a preference for A and disfavor for G at the −1 site and disfavor for C at the +1 site. Flanking sequence preferences were similar for TET1 and TET2 and also for 5mC and 5hmC substrates, but in general stronger effects were observed with TET2. Enhanced flanking sequence preferences were also observed at non-CpG sites together with profound effects of flanking sequences on the specificity of TET2. Strikingly, these preferences are reflected in genome-wide patterns of 5hmC in human cells and patterns of changes of DNA methylation after TET knock-out indicating that the flanking sequence preferences of TET enzymes discovered here represent an important parameter that influences genomic DNA modification patterns, which acts in combination with other processes like CpG site recognition and locus-specific targeting of the TET enzymes.

## Results

Since their discovery in 2009[4,5], many biochemical and structural studies have provided important insights into the function of TET enzymes and the mechanisms of DNA demethylation[13,17]. However, the influence of flanking DNA sequences on the activity of TET enzymes and their CpG recognition has not yet been systematically studied. We conducted a Deep Enzymology[30] study to investigate the effects of sequences flanking the target site on the activity of TET1 and TET2 using substrate libraries containing 5mC and 5hmC in CpG and non-CpG target sites embedded in a random flanking sequence context (Supplementary Fig. 1a). 5mC and 5hmC oxidation kinetics were conducted on the substrate mixtures and the oxidation state of the target bases and the corresponding flanking sequences of individual product molecules were determined by bisulfite conversion coupled to NGS (Fig. 1a and Supplementary Fig. 1). By this approach, the direct effect of the flanking sequence on enzymatic properties could be determined providing detailed insights into the influence of the substrate sequence on TET enzyme activity and specificity.

**Flanking preferences of TET1 and TET2 on CpG sites**. For our studies, we used the catalytic domains of murine TET1[32] and two versions of murine TET2 which both have flexible protein parts replaced by a linker as described for the human enzyme[20]. We prepared two libraries of substrates containing a hemimethylated (mCpG) or hemihydroxymethylated (hmCpG) CpG site in a randomized sequence context (Supplementary Fig. 1a). The substrate libraries were incubated with TET1 and both versions of TET2. Bisulfite conversion was used to detect the cytosine base oxidation (Supplementary Fig. 1b). In this reaction, unmodified cytosine, 5fC, and 5caC are deaminated, while 5mC and 5hmC are not converted[33] (Supplementary Fig. 1c). Hence, 5fC and 5caC are detected as thymine in subsequent DNA sequencing reactions, while 5mC and 5hmC yield cytosine. Therefore, the oxidation step from 5hmC to 5fC is observed, while the steps from 5mC to 5hmC and 5fC to 5caC are not visible. To obtain information about the 5mC to 5hmC step as well, we conducted

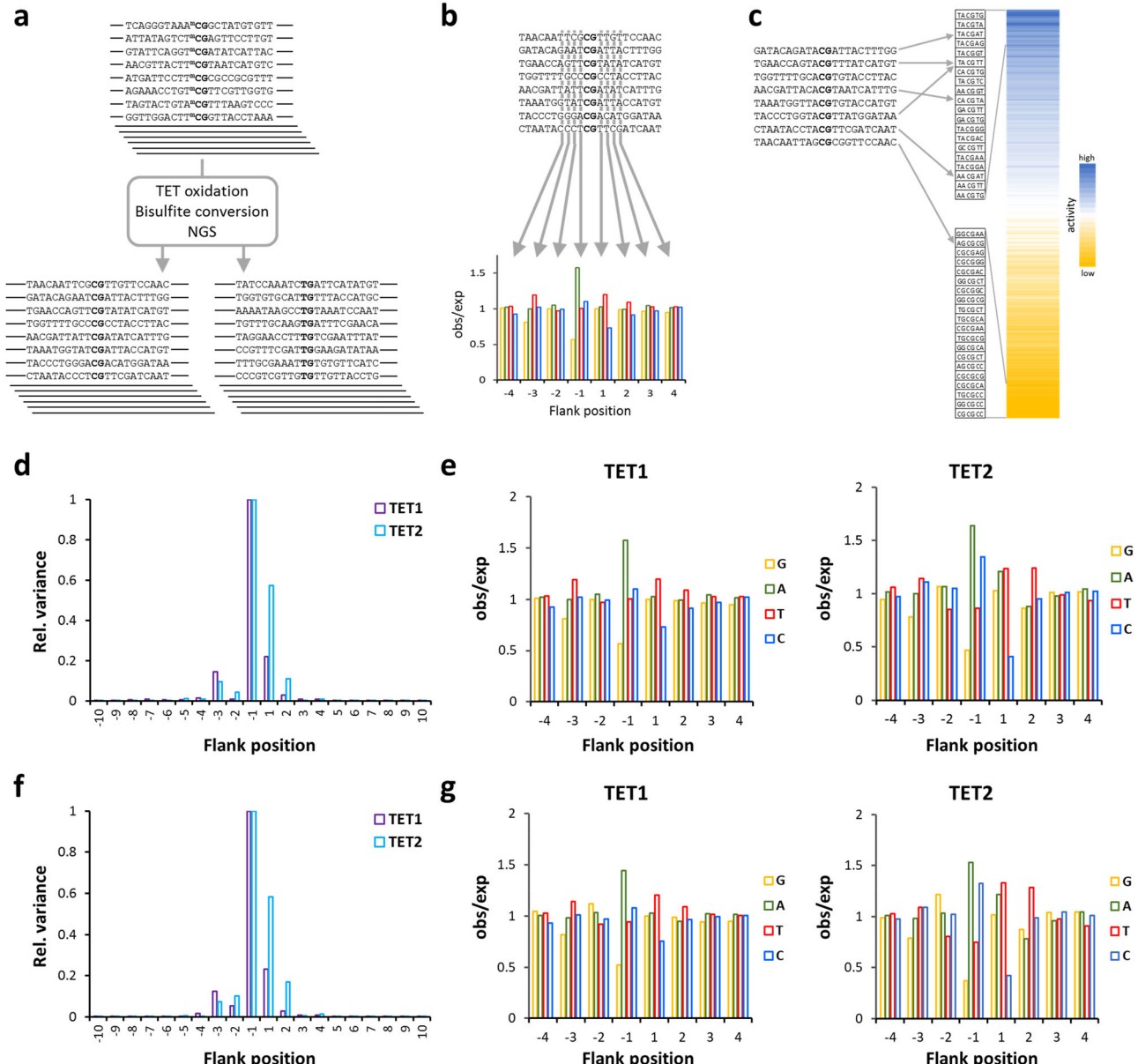

**Fig. 1 Global analysis of flanking sequence effects on mCpG and hmCpG oxidation by TET1 and TET2. a** Principle of the Deep enzymology approach. Libraries of DNA sequences containing one target cytosine (here mCpG) in a randomized context of ten nucleotides on either side are oxidized by TET enzymes and subjected to bisulfite conversion. By NGS the oxidation state and individual DNA sequences of single product molecules are determined (for details cf. Supplementary Fig. 1). **b** Principle of the first step of the data analysis. At each flanking position, the enrichment and depletion of individual bases in the methylated product pool is determined and expressed as observed/expected (obs/exp) values. **c** Principle of the second step of the data analysis. Methylation levels are averaged for all 256 NNCGNN sites, thereby revealing combinatorial flanking sequence effects. **d** Position-specific variance of oxidation of mCpG substrates. For the averaged TET1 and TET2 data sets, the sum of the (obs/exp -1)$^2$ values for G, A, T, and C were determined for all −10 to +10 flank positions and plotted after scaling to the largest value. The data show that the −3 to +2 flank positions have the largest influence on the methylation rate. **e** Average mCpG oxidation levels of substrates containing specific bases at −4 to +4 flank positions by TET1 and TET2. Oxidation levels are given as observed/expected (obs/exp) values. **f, g** Same as **d** and **e**, but referring to hmCpG substrates.

oxidation studies with 5mC and 5hmC substrates. NGS sequencing then allows determining the oxidation states of individual DNA molecules together with their specific flanking sequence (Supplementary Table 1). No-enzyme controls were conducted with the mCpG and hmCpG substrate libraries confirming the absence of conversion artefacts after the bisulfite treatment (Supplementary Table 2). In the sequences obtained by NGS from independent reactions for each pair of enzyme and substrate, we averaged the oxidation levels of mCpG and hmCpG substrates in all NNCGNN sequence contexts. With both substrates, we

observed very high correlations of the sequence preferences in the experimental repeats (Supplementary Figs. 2a, 3a). After the combination of the individual repeats, more than 50,000 NGS reads were available for each enzyme and substrate (Supplementary Table 1). In the averaged data sets, we observed very high similarity in the sequence preferences of both TET2 versions for mCpG (Supplementary Fig. 2b, c) and hmCpG (Supplementary Fig. 3b, c). As we obtained higher enzymatic activities with the TET2 version 2, all later reactions were conducted with this construct, which will be designated TET2 from here on.

**Analysis of TET1 and TET2 flanking preferences**. To analyze the effects of individual bases at the different flank positions on TET1 and TET2 activity, we used the averaged data and determined the relative enrichment or depletion of each base at each flanking position in the pool of oxidized substrates and expressed it as observed/expected ratio (Fig. 1a, b). This value directly indicates if a particular base at the corresponding flank position enhances or reduces the catalytic activity. We first collected the variances of obs/exp values determined for each flank position, indicating that the −1 site has the strongest effects, followed by +1 and the −3, −2, and (in case of TET2) +2 sites, which have a more moderate effect on activity (Fig. 1d, f). Our data show that for both enzymes and both substrates, an A is strongly preferred at the −1 site and G is strongly disfavored (Fig. 1e, g). At the +1 site, C is generally disfavored and at the +2 site, TET2 prefers T.

To analyze the combined effects of flanking positions in more detail, the 5mC and 5hmC oxidation activities were averaged for all 256 NNCGNN sites (Fig. 1c) and each of them fitted to a monoexponential reaction progress curve to determine average rate constants of oxidation of the substrates with the corresponding flanking sequence (Supplementary Fig. 4). The data revealed strong differences in the average oxidation rates of target bases embedded in different flanking sequence contexts (Fig. 2a), viz. a 22-fold ratio in the oxidation rates of the best and worst 5mC substrate for TET1, 8.5-fold for TET1 on 5hmC, 69-fold for TET2 on 5mC, and 25-fold for TET2 on 5hmC. These results indicate that flanking sequence preferences of TET2 are stronger than those of TET1. Heatmaps (Fig. 2a) and correlation factors (Fig. 2b) revealed very high similarities of the flanking sequence preferences of TET1 and TET2 for the oxidation of 5mC and 5hmC substrates. Next, we inspected the groups of most preferred and most disfavored flanking sequences for both enzymes and both substrates and prepared Weblogos for visualization (Fig. 2c). In addition to the features already mentioned above, this analysis revealed that TET1 prefers a TA dinucleotide at the −2/−1 site, particularly with 5mC substrates. Strikingly, TG at the same place is disfavored in particular by TET2. Moreover, a TT dinucleotide is preferred at the +1/+2 site, particularly by TET2 and with 5hmC substrates.

As described above, in our bisulfite assay the conversion of 5hmC to 5fC is detected. Hence, the oxidation of 5hmC substrates is directly visible. In contrast, on 5mC substrates, the oxidation of 5mC to 5hmC must occur before the next oxidation step to 5fC can occur. Hence, in the reaction with the 5mC substrates, the combined effects of both oxidation steps are detected. The high similarity of the flanking sequence preferences observed with 5mC and 5hmC substrates suggests that flanking sequence preferences of both oxidation steps are similar, which is structurally reasonable, as the conformations of the TET2 structures with bound 5mC and 5hmC substrates are very similar[22]. However, we noticed that the overall preferences were more pronounced with 5mC substrates, which can be explained because in this case, two oxidation steps are necessary before the conversion is detectable and each of the individual steps contributes to the flanking sequence preferences leading to an amplification of the effects.

Finally, we compared the NNCGNN flanking sequence preferences of TET1 and TET2 with the previously determined flanking sequence methylation preferences of DNMT1[28] and DNMT3A and DNMT3B[27] (Supplementary Fig. 5). The data illustrate the high similarity of flanking sequence preference profiles of TET1 and TET2. Strikingly, however, the overall preferences of TET enzymes, DNMT1, DNMT3A, and DNMT3B enzymes are quite different, which reflects the differences in the DNA binding and recognition process in all these enzymes.

**Validation of TET1 and TET2 flanking sequence preferences by LC-MS**. To confirm the flanking sequence preferences observed in the previous sections with an unrelated technology, four synthetic double-stranded oligonucleotide substrates were used which contain 5mC or 5hmC in a favorable (TACGTA, rank 2 of 256 for TET1 and 29 of 256 for TET2, where low numbers indicate high preference) or unfavorable sequence context (CGCGCC, rank 256 for TET1, rank 244 for TET2) (Supplementary Table 4). The substrates contained one 5mC or 5hmC in one DNA strand and were oxidized by TET1 and TET2. Then, the substrates were hydrolyzed to nucleosides and the progress of the TET reaction was observed by liquid chromatography coupled to mass spectrometry (LC-MS) detecting the characteristic fragmentation of nucleosides at the glycosidic bond (Supplementary Fig. 6a). Using standards, the elution profiles were established and calibration curves determined (Supplementary Fig. 6b, c). Using the reaction progress curves, the corresponding rate constants were determined by numerical integration (Fig. 3a).

Initial attempts to fit the experimental data to sets of differential equations revealed that a model only including three rate constants ($k_1$, $k_2$, and $k_3$) for the stepwise conversion of 5mC to 5hmC, 5hmC to 5fC, and finally 5fC to 5caC could not describe the LC-MS reaction progress curves observed with the preferred substrate (Supplementary Fig. 7). Therefore, a processive reaction step directly converting 5mC to 5fC ($k_{12}$) was included into the model (Fig. 3b), which led to a strong improvement of the fit. This observation suggests that under our reaction conditions TET1 and TET2 at least partially catalyze the oxidation of substrates in a processive mechanism. Based on our data, it cannot be excluded that reactions fitted with consecutive rate constants also followed a processive mechanism.

Two independent reactions were conducted with both enzymes using the 5mC and 5hmC substrates. In each case, the preferred and disfavored substrates were used side-by-side and the data analyzed together resulting in small to medium-sized error bars (Fig. 3c). The data reveal the expected order of catalytic activities with $k_1 > k_2 > k_3$. Moreover, in each case, the rate constants determined with the preferred substrate were larger than the ones determined with the disfavored substrates, although to a variable degree between ~3-fold and more than 50-fold. Of note, the average of the six ratios of rate constants on preferred over disfavored substrates was larger for TET2 (47.5) than for TET1 (14.2), which is in agreement with the Deep Enzymology data also demonstrating more pronounced flanking sequence preferences of TET2.

**Sequence preferences in readout of 5hmC**. Having identified hmCpG sites in favorable and disfavored sequence context for TET enzymes, we were interested to find out, if 5hmC readout is also affected by the sequence context. To investigate this question, we resorted to UHRF2[14] that binds 5hmC with a specific pocket within its SRA domain[15]. Using gel shift experiments with synthetic double-stranded 30mer oligonucleotides containing one unmodified CpG, one hemimethylated (mCG/CG), and one hemihydroxymethylated (hmCG/CG) CpG site, we studied DNA binding of the purified human UHRF2 SRA domain and observed a preferential interaction with the 5hmC substrate (Fig. 3d), as reported previously[15]. Next, we conducted gel shift experiments with the two different 30mers containing the single hmCpG target site in different flanking contexts (Fig. 3e). These experiments revealed a strong effect of the flanking context on DNA binding of UHRF2 indicating that 5hmC readout is also flanking sequence-dependent.

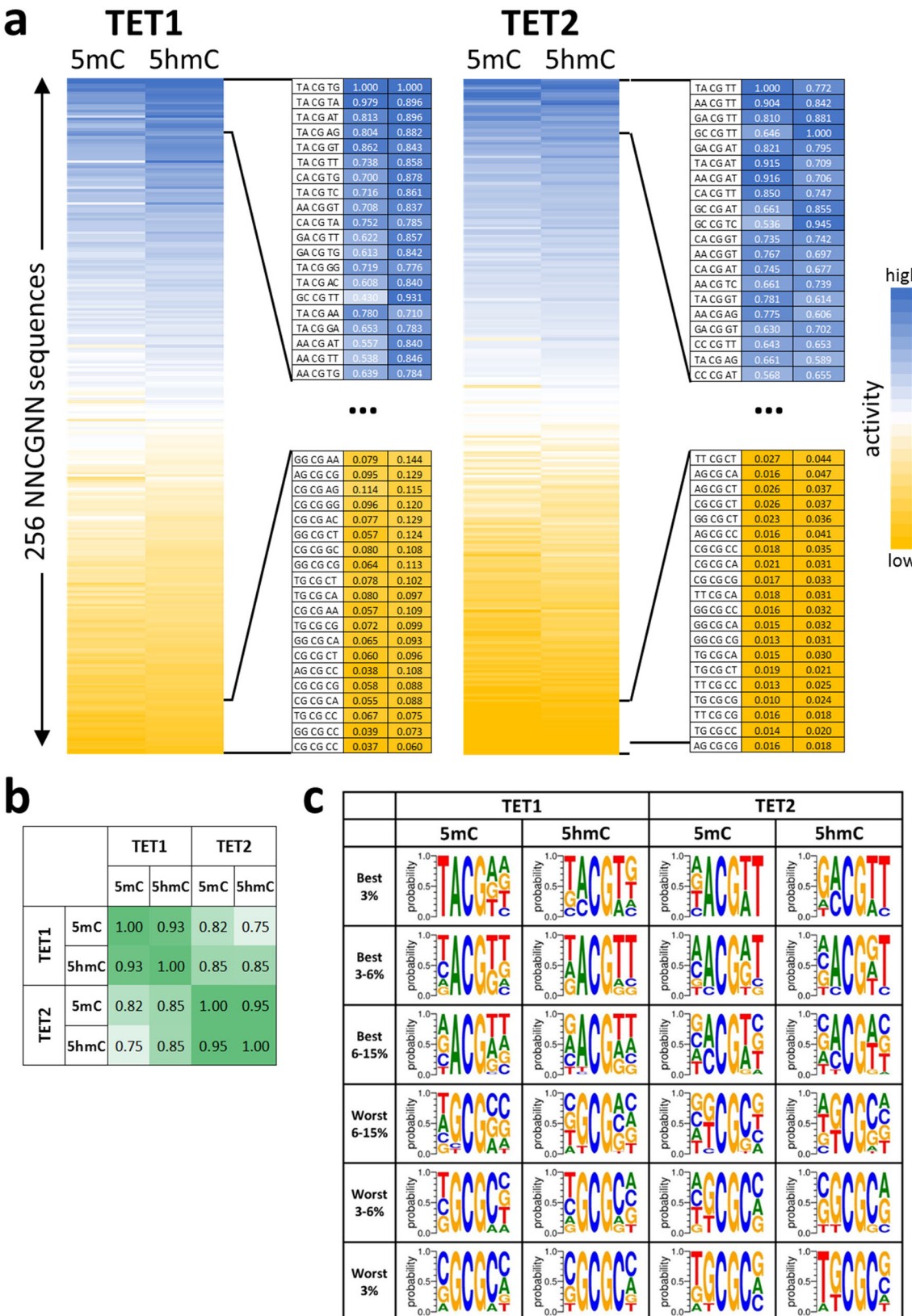

**Fig. 2 Flanking sequence effects on the oxidation of mCpG and hmCpG substrates by TET1 and TET2. a** Heatmap of the average activities of TET1 and TET2 at NNmCpGNN and NNhmCpGNN sites sorted by the average activity. The enlargements show the activity and sequence of the most preferred and disfavored flanking sequences. **b** Pairwise Pearson correlation factors of the flank profiles shown in panel **a**. All data were compiled in Supplementary Data 1. **c** Weblogos of the enrichment of bases at the different flank positions in subsets of the most preferred and disfavored NNCGNN flanking sequences. Weblogos were prepared using WebLogo 3 (http://weblogo.threeplusone.com/).

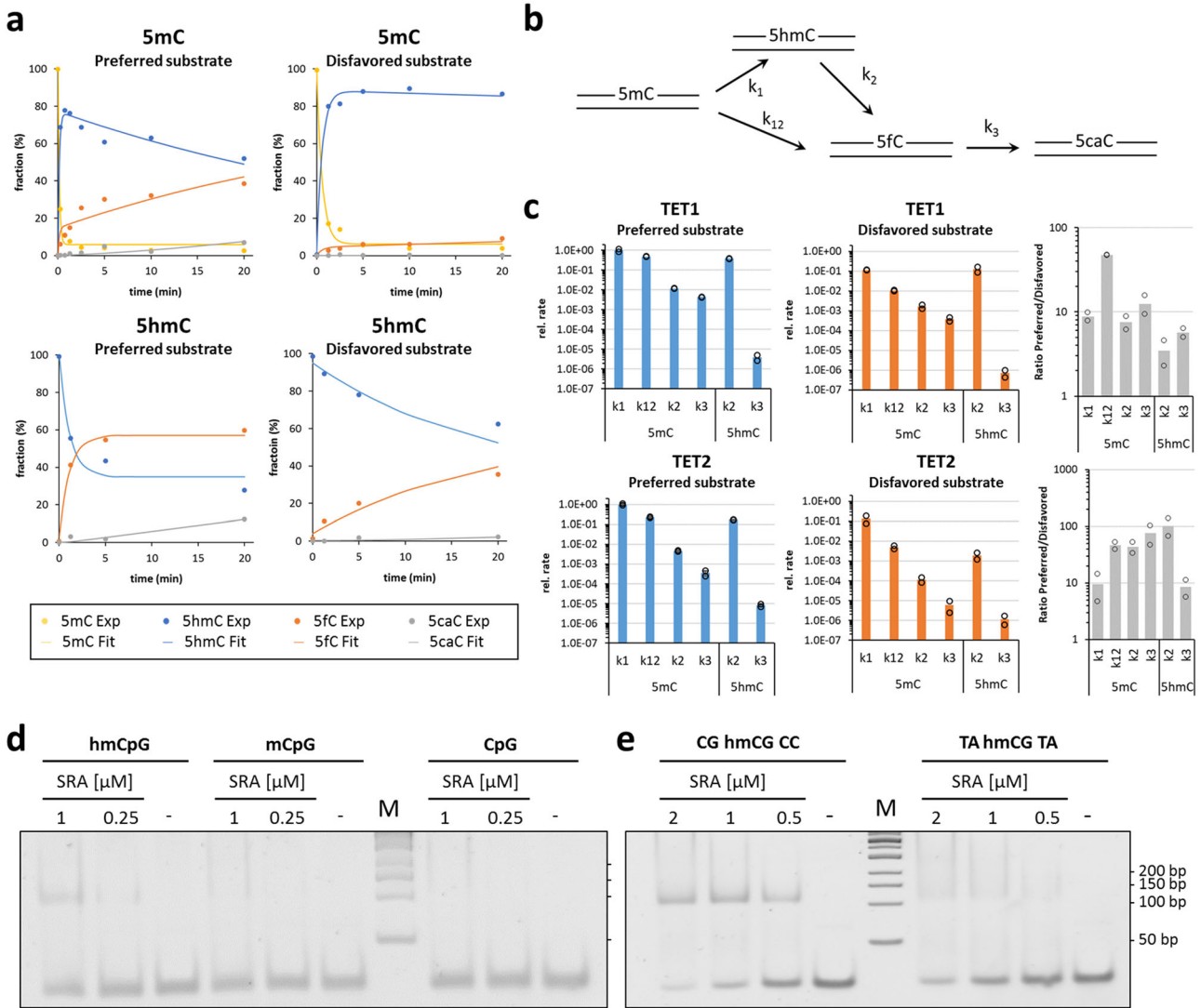

**Fig. 3 Biochemical investigation of the flanking effect of TET1 and TET2 oxidation kinetics and 5hmC binding by UHRF2. a** Example of oxidation kinetics of synthetic double-stranded 30mer oligonucleotides containing one hemimethylated (5mC) or hemihydroxymethylated (5hmC) CpG site in different flanking context by TET2. Product appearance was detected by LC-MS. Enzyme concentrations of TET1 and TET2 were 1.6/2.0 μM for the favored substrates and 3.2/8 μM for the disfavored substrates. **b** Kinetic model used to analyze the data. **c** Summary of rate constants of oxidation of 5mC and 5hmC substrates by TET1 and TET2. Rates are given as relative values considering that in the reactions with the disfavored substrates more enzyme was used. Shown are averages and data points of two independent repeats. **d** Gel shift experiments with purified UHRF2 SRA domain and synthetic double-stranded 30mer oligonucleotides (0.5 μM) containing one hemihydroxymethylated CpG site (hmCpG), one hemimethylated (mCpG), and one unmodified CpG site in CGCGCC context. **e** Gel shift experiments with purified UHRF2 SRA domain and synthetic double-stranded 30mer oligonucleotides (0.5 μM) containing one hemihydroxymethylated CpG site in different flanking context.

**Sequence preferences in the oxidation of non-mCpG sites**. To investigate flanking sequence preferences of TET1 and TET2 for the oxidation of 5mC in a non-CpG context, substrate libraries were generated which contained an mCpX site in a random flank context. Two separate reactions were conducted with TET1 and TET2 and the average oxidation levels of mCpX sites were determined (Supplementary Table 1). The −4 to +4 flanking sequence preferences observed in both experimental repeats were highly correlated (Supplementary Fig. 8). Correlation factors for mCpT oxidation were slightly lower, due to the low overall product levels, which led to increased fluctuations. After averaging the data from both repeats, overall activities were determined for 5mC oxidation at CpG and non-CpG sites (Fig. 4a). With both enzymes, activities were highest at CpG sites. For TET1 the next best target site was mCpA, followed by mCpC and mCpT (mCpG > mCpA > mCpC > mCpT). In case of TET2, the preferences for mCpC and

mCpA were swapped (mCpG > mCpC > mCpA > mCpT). As mCpT oxidation levels were quite low (~5% for TET1 and ~2% for TET2), it was not included in the further analyses.

Comparison of the flanking sequence preferences at the −4 to +4 flank positions (Fig. 4b) showed similar overall profiles for mCpG, mCpA, and mCpC, in line with the high pairwise correlation factors. Interestingly, the flanking sequence-dependent fluctuations of oxidation rates were higher on mCpA and mCpC sites than on CpG sites, suggesting that on the less active substrate, the quality of the fit of the flanking sequence to the enzyme's preferences becomes more important. A similar observation had been made previously with DNMT3B[29]. Moreover, we observe that the CpG recognition is modulated by the flanking base pairs, as indicated by the strong preference of TET1 and TET2 for mCpA oxidation with a T at the +1 site. In contrast, mCpC oxidation by TET2 is preferred with A(+1).

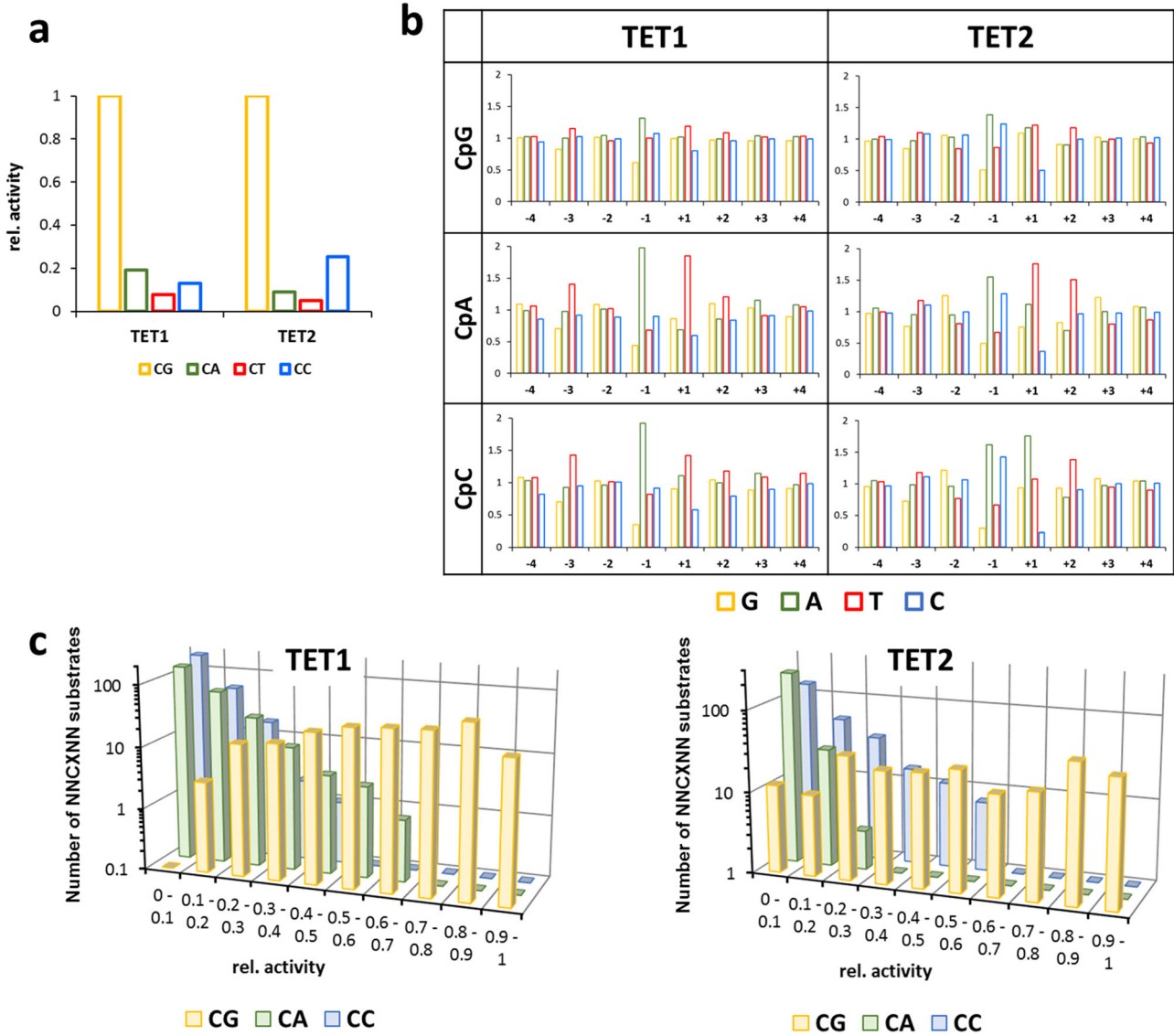

**Fig. 4 Flanking sequence effects on the oxidation of mCpX substrates by TET1 and TET2. a** Relative oxidation of different CpX (X=G, A, T, or C) substrates. **b** Average oxidation levels of mCpX substrates containing specific bases at −4 to +4 flank positions. Oxidation levels are given as observed/expected values. **c** Frequency of NNCXNN substrates within defined ranges of relative activities showing the overlap of activity ranges of preferred non-CpG and disfavored CpG sites.

As described so far, our analysis averaged over all flanking sites revealed mCpG as the best substrate for TET1 and TET2, with mCpA and mCpC as the second-best target sites for TET1 and TET2, respectively, but the enzymatic activities at each of the sites vary strongly depending on the flanking context. For a more detailed analysis, the relative activities of oxidation of all NNmCpGNN, NNmCpANN, and NNmCpCNN substrates were determined (Fig. 4c). Since these reactions were conducted in competition in one reaction tube, relative activities with the different CpX substrates can be directly compared. A frequency plot of the activity ranges of different substrates revealed a broad overlap in activity between mCpG and non-CpG substrates. For example, 13–38% of the CpG and CpA substrates were in the 20–40% activity range of TET1, and, similarly, 20–30% of the CpG and CpC substrates were in the corresponding activity range of TET2. This means that there is large group of preferred CpA or CpC substrates, whose oxidation is comparable or even faster than the oxidation of the corresponding group of disfavored CpG substrates. This analysis indicates that TET enzymes are well

suited for the generation of 5hmC at non-CpG sites, TET1 particularly at CpA and TET2 particularly at CpC sites. Similarly for both enzymes the activity ranges of CpC and CpA activity were largely overlapping.

**Correlation of flanking sequence preferences of TET enzymes with genome-wide patterns of 5mC and 5hmC.** To analyze the cellular effects of TET flanking sequence preferences, global DNA methylation, and hydroxymethylation patterns in human lung and liver cells were taken from published whole-genome data[10] and averaged 5hmC levels of CpG sites in different flanking sequence context were determined. In addition, changes of DNA methylation after knock-out of TET1 or TET2 in mouse embryonic stem cells were taken from published reduced representation genome bisulfite data[34] and the average increase in methylation of CpG sites after TET KO (Δm) was calculated for different flanking sequence contexts. As the TET enzymes showed the strongest flanking sequence effects at the −1 and +1 flanking

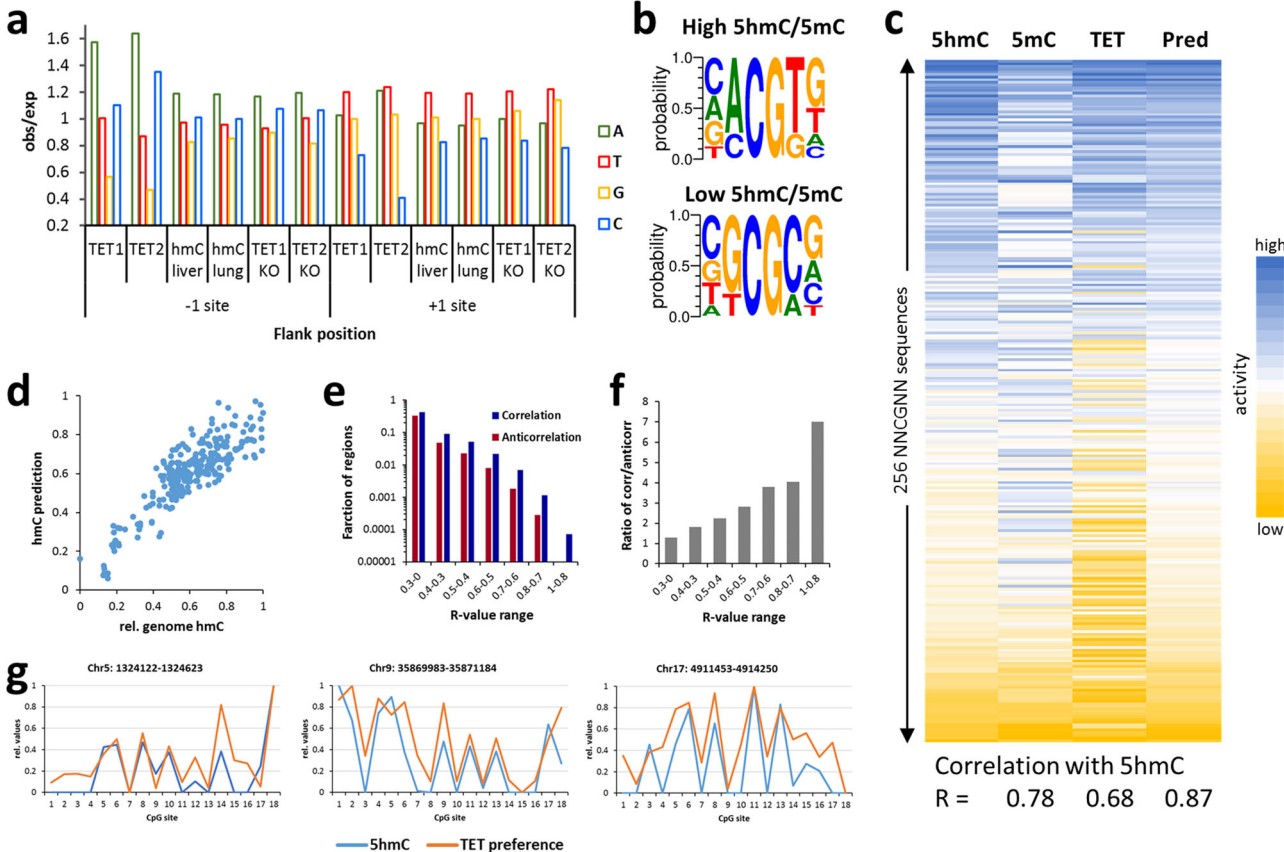

**Fig. 5 Correlation of genomic DNA modification patterns with flanking sequence preferences of TET1 and TET2. a** Preferences of TET1 and TET2 for bases at the −1 and +1 flank site compared with the enrichment or depletion of bases at these flanking positions in genomic 5hmC pattern[10] and at sites associated with gain of genomic 5mC content after TET1 or TET2 knock-out (KO) in mouse ES cells[34]. **b** Average genomic 5hmC levels were determined for all NNCGNN sites and compared with average genomic 5mC pattern at the same sites[28,35]. Shown are Weblogos of the sites with the highest and lowest ratios of 5hmC and 5mC contents. **c** Heatmaps of averaged genomic 5hmC levels (5hmC), genomic 5mC levels (5mC), averaged TET flanking sequence preferences (TET), and the prediction of 5hmC levels based on the combination of 5mC levels and TET flanking preferences (Pred). **d** Scatter plot of genomic 5hmC levels and its prediction from panel **c**. **e** The Pearson R-value was determined for the correlation of genomic 5hmC levels and average TET1 and TET2 NNCGNN preferences for regions of 18 consecutive CpG sites sliding over the genome. Frequency plot of the distribution of R-values among all regions. Positive R-values (dark blue bars) indicating a correlation of TET preferences and 5hmC patterns were observed much more frequently than negative R-values (dark red bars). **f** Ratio of the fractions of regions with positive and negative R-values shown in panel **e** in the different R-value ranges. **g** Example regions selected from arbitrary parts of different chromosomes showing the correlation of local 5hmC levels and average TET1 and TET2 NNCGNN preferences. 5hmC level and TET preferences were normalized to the highest and lowest values.

sites, we focused our initial analysis on NCGN sites. As shown in Fig. 5a, the sequence preferences of the TET enzymes are in very good agreement with the enrichment and depletion of bases at genomic 5hmC sites and with genomic Δm-values, because in all cases A is preferred and G is disfavored at the −1 site while T is preferred and C disfavored at the +1 site.

Assuming that genomic 5hmC patterns must be determined by 5mC levels (the substrate for TET enzymes) and the flanking sequence preferences of the TET enzymes, the ratio of 5hmC and 5mC levels should most closely correspond to the TET enzyme flanking sequence preferences. To test this presumption, the published genome-wide 5hmC levels[10] were used to determine average 5hmC levels for all NNCGNN sites which were compared with genomic 5mC pattern[35] that were also averaged at the NNCGNN sites[28]. For this, the 5hmC data from lung and liver cells were averaged based on their very high overall correlation (Supplementary Fig. 9). Ratios of 5hmC and 5mC contents were calculated and the sites with the highest and lowest 5hmC/5mC ratios were used to prepare Weblogos (Fig. 5b). Strikingly, sites with a high genomic 5hmC/5mC ratio are preferentially found within an A(−1) and T(+1) context, while low genomic 5hmC/

5mC ratio is associated with G(−1) and C(+1) flanks. These flanking preferences are in perfect overall agreement with the preferences of TET1 and TET2 at the −1 and +1 flanks, suggesting that high or low 5hmC/5mC levels are caused by high or low TET activity at these specific sites, respectively. We next calculated the correlation between 5hmC levels in all NNCGNN sites with the corresponding 5mC levels and averaged TET flanking sequence preferences. Indeed, we observed a clear correlation in both comparisons (Fig. 5c). To simulate a setting in which both properties, 5mC distribution and TET enzyme flanking sequence preferences, together determine 5hmC levels, we combined the 5mC and TET activity patterns and tested the correlations of these mixed patterns with the 5hmC levels. Strikingly, a combination of 36% of TET activity and 64% 5mC levels could describe the genomic 5hmC patterns very well (Fig. 5d) and better than both of the individual distributions alone.

After having observed strong correlations of TET flanking sequence preferences and aggregated genomic 5hmC levels in defined flanking contexts (NCGN or NNCGNN), we were interested to find out if local 5hmC patterns are also determined

by TET flanking preferences. To this end, the Pearson correlation factors of 5hmC level and average TET preferences in the NNCGNN context were determined for regions of 18 consecutive CpG sites sliding over the genome. Strikingly, positive correlation factors were observed much more frequently than negative correlation in particular for higher R-values (Fig. 5e, f). This result indicates that local 5hmC patterns are also influenced by TET flanking preferences as also illustrated in example regions (Fig. 5g).

## Discussion
In the last 2 years novel experimental approaches have shown pronounced effects of flanking sequences on the CpG recognition of DNMT3A, DNMT3B, and DNMT1[27–29], leading to differences in the substrate preferences of both DNMT3 paralogs[27] or DNMT3A and its somatic cancer mutant R882H[26]. Mechanistically, flanking sequences can affect indirect DNA shape readout[26–28], but also direct DNA contacts and alternative contact networks between the enzymes and different flanking sequences were found[27,29]. In DNMT1, striking differences in the mechanism of DNA rearrangement after base flipping were observed in different flanking contexts that were directly coupled to enzyme activity[28]. Here we have applied a Deep Enzymology approach for the analysis of the detailed flanking sequence preferences of TET1 and TET2 and documented more than 50-fold differences in oxidation rates of 5mC residues placed in different NNCGNN flanking contexts. The flanking sequence preferences of the TET enzyme were detected in our study using bisulfite conversion coupled to NGS and validated in exemplary cases by LC coupled mass spectrometry. Of note, flanking sequence preferences of TET enzymes were not detectable in previous studies, which investigated TET2 activity only with a small number of different substrates using conventional enzyme assays[17,22]. Our experiments were conducted using substrates with hemimethylated CpG sites in a randomized flanking sequence context. It will be an interesting topic for future research to investigate the effect of the modification state of the CpG cytosine in the non-target strand on the DNA recognition and flanking sequence preferences of TET enzymes.

To rationalize our findings, we inspected available structures of TET enzymes in different −1/+1 flanking contexts, viz. TET2 structures in a T/A (5D9Y)[22] and C/G contexts (5DEU and 4NM6)[20,22] and the structure of *Naegleria* Tet-like dioxygenase Ng-TET in A/C context (4TL5)[21]. We observed activity effects for the sequence ranging from the −3 to +2 flanking positions, which perfectly agrees with the range of base pairs that are in close contact with the protein in the TET2 structures (Fig. 6a). Comparison of the two TET2 structures reveals that in 5DEU Arg1302 contacts the CG base pair at the +1 site by a direct hydrogen bond to G(+1′) in the minor groove (Fig. 6b). In contrast, only a water-mediated contact is formed in 5D9Y, which carries a TA base pair at the +1 site. Strikingly, the CG(+1) base pair is shifted by 1.5 Å in the direction of the helix axis in 5DEU, which may be related to the direct sequence contact. This effect may increase the stacking of the target C base in the helix, making base flipping more difficult which could explain the disfavor for C at the +1 flank position. Finally, we notice that in Ng-TET the G′ residue (the Watson/Crick partner base of the target 5mC) is not rotated out of the double helix, while both TET2 structures do show the rotation of this base out of the helix (Fig. 6c). While one cannot exclude enzyme-specific effects, it is striking to note that in the Ng-TET structure an AT base pair is present at the −1 site, which is the most preferred base pair at this site, while the TET2 complex structures contain CG and TA pairs. One may speculate that AT(−1) stabilizes the intrahelical position of the G′ and the

absence of G′ flipping may explain the particular preference of TET enzymes for A(−1). A similar observation has been made previously with DNMT1, where we also showed that binding sites showing strong conformational changes in the complex are characterized by low methylation activity and vice versa[28]. In this respect it is interesting to note, that the rotation of G′ is more pronounced in the 5D9Y complex with TA(−1) base pair than in 5DEU with CG(−1) base pair, which also fits the flanking preference of TET2 with T(−1) > C(−1).

We further observed that CpG recognition of TET enzymes is modulated by the flanking base pairs, because TET2 prefers mCpA oxidation with T(+1) while mCpC oxidation is preferred with A(+1). This finding could explain a discrepancy in previous reports regarding the second-best mCpX target site of TET2 following mCpG. Hu et al. (2013) reported a preference for mCpC[20] while DeNizio et al. (2021) reported mCpA[23]. Indeed, DeNizio et al. used a substrate with a T at the +1 site, which is preferred in a mCpA context, while the substrate of Hu et al. carried a G at this site that is equally preferred in both contexts.

Genomic DNA modification patterns represent a dynamic steady-state determined by the local activities of DNA methyltransferases and demethylating processes[3]. It is well established that genome targeting of DNMTs and TETs depends on their interaction with other proteins like transcription factors, chromatin modifications, and non-coding RNAs[2,36,37]. However, after recruitment of a DNMT or TET enzyme to a genomic target locus, the DNMT or TET will modify the target residue with variable efficiency according to its sequence and flanking sequence preferences. At first instance, DNMTs and TETs preferentially interact with target cytosines in a CpG context. However, methylation at non-CpG sites has also been observed[38] and 5hmC has been found in non-CpG context as well[39–41], although its amounts are still under debate. Mellen et al. (2017) reported relatively high levels of non-CpG 5hmC preferentially at CpA sites in neurons[40], where hmCpA constitutes a potential MeCP2 binding site[42,43]. Our data suggest that TET1 is the main enzyme responsible for hmCpA generation, in agreement with the finding that TET1 is necessary for DNA demethylation in neurons[16] and TET1 deficiency in mice leads to neuronal phenotypes[44,45].

In this work, we demonstrate that TET enzymes oxidize target sites with up to 70-fold differences in activity depending on the flanking sequence context. Based on this, we document strong differences in average genomic 5hmC levels at CpG sites with different flanking sequences, which are highly correlated with the flanking sequence preferences of the TET enzymes, as well as local 5hmC patterns that recapitulate TET flanking preferences. These findings indicate that flanking sequence preferences represent an important, so far not considered parameter that influences genomic DNA modification patterns. Flanking sequence preferences act in combination with the other already known processes like the CpG specificity and locus-specific targeting of TET enzymes and their regulation by chromatin modifications at the target regions and by posttranslational modifications or interacting proteins. Our data indicate that after recruiting a TET enzyme to a target locus and activating it by PTMs and binding of complex partners, the local CpG site-specific activity depends on the flanking sequence preferences of the catalytic domain. Hence, the overall genomic 5mC and 5hmC patterns are defined by these different processes acting together. Strikingly, DNMTs follow the same principle mechanism, because they also modify CpG target sites with variable efficiencies according to flanking sequence preferences and this leads to differences in genome-wide average 5mC content of CpG sites with different flanks[27–29]. It needs to be noted that the different DNA sequence-dependent processes determining the activity of

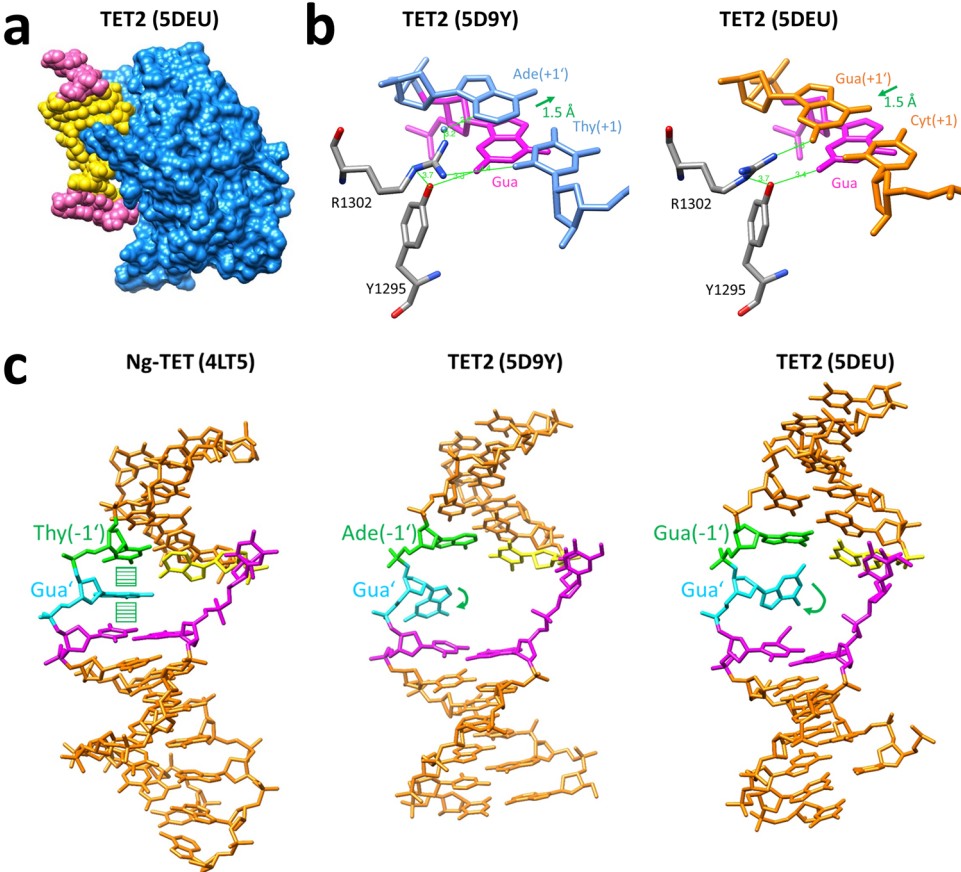

**Fig. 6 Structural details of TET enzyme-DNA complexes. a** Structure of TET2 with the −3 to +2 flanking region of the DNA colored in yellow. **b** Comparison of two TET2 structures revealing altered positions of the +1 flank base pair (shown in blue and orange) and changes in the hydrogen bonding network (green lines) involved in the recognition of Gua' (shown in purple). **c** Comparison of three TET structures revealing different conformations of the Gua' (shown in cyan) depending on the −1 flanking base pair (shown in green and yellow). The other residues of the CpG site are shown in purple.

DNMT and TET enzymes are connected because CpG recognition is attenuated by the flanking sequence context of the CpG and non-CpG sites particularly for the flank position +1 as shown here for TET enzymes and previously for DNMT3B[29]. However, the overall flanking sequence preferences of TET enzymes, DNMT3 enzymes, and DNMT1 are completely different. Future work needs to address the details of the crosstalk of all these processes, which together determine global DNA modification patterns and epigenetic information transfer.

## Methods

**Cloning, protein overexpression, and purification.** Plasmids encoding the His-tagged mouse TET1 catalytic domain (amino acid 1367–2057 of XP_006513930.1)[32] and mouse TET2 full-length[46] were purchased from Addgene (plasmid #81053, and 89735). The catalytic domain of TET2 based on Ito et al.[6] was cloned into the TET1 expression vector. To increase the expression yield, amino acids 1401–1764 in the TET2 sequence were deleted and replaced by a 15 amino acid linker (GGGGSG GGGSGGGGS) as previously described for the human enzyme[20] yielding TET2 version 1 (amino acid 915–1400-linker-1765–1920 of XP_006501349.1). In addition, the catalytic domain of another TET2 isoform was prepared through the insertion of serine by site-directed mutagenesis yielding TET2 version 2 (amino acids 915–1401-linker-1765–1920 of NP_001333665.1). The TET enzyme constructs are schematically shown in Supplementary Fig. 10.

For overexpression, all three TET constructs were transformed into BL21 (DE3) Codon+ RIL *E. coli* cells (Stratagene). The cells were grown in LB medium supplemented with trace metals[47] and streptomycin until an $A^{600,nm}$ of 0.6 was reached, and protein expression was induced for 12–14 h at 20 °C by the addition of 0.5 mM isopropyl-1-thio-D-galactopyranoside (Roth). Purification was carried out similarly as described in ref. [48]. In brief, harvested cells were washed once with 1X STE (10 mM Tris-HCl pH 8, 100 mM NaCl, 1 mM EDTA) buffer, the pellet was then resuspended in sonication/wash buffer (50 mM HEPES pH 6.8, 500 mM NaCl, 1 mM DTT, 35 mM imidazole, 10 mM alpha-ketoglutarate, and 10% glycerol)

supplemented with Protease Inhibitor cocktail (final 300 μM AEBSF-HCl, 3 μM Pepstatin A, 0.12 μM Aprotinin, 15 μM Bestatin, 4.5 μM E-64, 6.7 μM Leupeptin) and 0.2 mM PMSF (Sigma) and lysed by sonication (15 cycles, 15 s with 30% power, 45 s off). The lysate was centrifuged for 1 h at 47,400 × g and 4 °C, batch binding to Ni-NTA beads (Genaxxon) was performed with constant rotation for 1 h and after washing with sonication/wash buffer the proteins were eluted (50 mM HEPES pH 6.8, 500 mM NaCl, 1 mM DTT, 300 mM imidazole, 10 mM alpha-ketoglutarate, and 10% glycerol) and dialyzed against dialysis buffer (50 mM HEPES pH 6.8, 300 mM NaCl, 1 mM DTT, 10 mM alpha-ketoglutarate, and 10% glycerol). Aliquots of the dialyzed TET proteins were frozen in liquid nitrogen and stored at −80 °C. The quality of the proteins was verified via Coomassie-stained SDS polyacrylamide gels and the concentration was determined based on $A^{280,nm}$.

**Deep Enzymology reactions with randomized substrates.** For analysis of flanking sequence preferences of the TET enzymes, a similar approach as described for DNMTs[27] was used. Briefly, single-stranded oligonucleotides (Supplementary Table 3) containing a methylated or hydroxymethylated CpG or CpH site flanked by ten randomized nucleotides on either side were obtained from IDT. Primer extension was performed with an extension primer to obtain the double-stranded DNA substrates, which were purified using a PCR clean-up kit (MACHEREY-NAGEL). A CpN substrate was prepared as a mixture of CpG and CpH in a 1:3 ratio. For the randomized hemihydroxymethylated substrate, the single-stranded oligo with the hydroxymethylated CpG site was purchased from IDT coupled to Desthiobiotin-TEG. Primer extension was conducted and the substrate was purified via Streptavidin beads (Dynabeads M-280, ThermoFisher Scientific) as described in the protocol of the supplier and eluted with a biotin solution (1 mM in Tris pH 8.8). The randomized double-stranded substrates (122 nM) were incubated with the purified TET enzyme at 37 °C for 45 min (CN context) or 1 h (CG context) using mixtures containing reaction buffer (50 mM HEPES pH 6.8, 100 mM NaCl, 1 mM DTT, 1 mM alpha-ketoglutarate, and 2 mM ascorbic acid), 100 μM ammonium iron(II) sulfate, using different enzyme concentrations and variable amounts of dialysis buffer to keep a fixed salt and glycerol concentration. Reactions were stopped by freezing in liquid nitrogen. Afterward, Proteinase K (NEB) treatment was used for enzyme inactivation for 1 h at 50 °C, followed by

purification with a PCR clean-up kit (MACHEREY-NAGEL). Afterward, the DNA was digested with the BsaI-HFv2 (NEB) and a hairpin was ligated using T4 DNA ligase (NEB). Bisulfite conversion was performed using the EZ DNA Methylation-Lightning kit (ZYMO).

**Library preparation for the Deep enzymology reactions and bioinformatics analysis**. Library preparation for Illumina NGS was conducted as described in ref. [27] using a two-step PCR approach. Unique combinations of barcode and index sequences (Supplementary Table 3) were introduced to distinguish different samples and experiments. For bioinformatic analysis of the NGS data sets, a local instance of a Galaxy server[49] was used. Sequence reads were trimmed with Trim Galore! (Galaxy Version 0.4.3.1, https://www.bioinformatics.babraham.ac.uk/projects/trim_galore/) keeping only the sequences with a quality score above 20 for further analysis, and filtered according to the expected DNA size using the Filter FASTQ tool[50]. Reconstitution of the original DNA sequence was executed based on the bisulfite converted upper and lower strands to determine the average modification state of the upper CpX site for each base at different flanking positions as well as for all NNCpXNN flanks. In this experiment, the methylated and hydroxymethylated states could be distinguished from formylated and carboxylated states. Pearson correlation factors were calculated with Excel using the correl function. In general, results of independent experimental repeats correlated closely (Supplementary Figs. 2, 3, and 8), therefore all repeats were averaged for downstream analysis.

**Combined kinetic fitting of NNCGNN substrates**. Reactions of individual substrates are assumed to be independent and reaction velocities are of first order with respect to the substrate concentrations. The time courses of the products are described as in Eq. 1:

$$z_i(t, k_i) = 1 - e^{-k_i t} \qquad (1)$$

For each substrate, the corresponding $k_i$ was determined by a least-squares approach. We wanted to exploit the particular power of competitive kinetics that relative turnovers of all substrates at a given point of the reaction progress curve are precisely known. This resulted in a coupled optimization problem for all substrates in one dataset, in which we optimized the oxidation rate constants $k_i$, $i = 1, ..., 256$ and the measurement times $t_j$, $j = 2, ..., n$ simultaneously. As this can only be done in arbitrary time units, we set $t_1 = 1$, resulting in Eq. 2:

$$\theta = (t_1 = 1, t_2, .., t_n, k_1, ..., k_{256}) = \underset{\theta}{\operatorname{argmin}} \sum_{j=1}^{n} \sum_{i=1}^{256} (y_i(t_j) - z_i(t_j, k_i))^2 \qquad (2)$$

Here, we minimize the sum of squared differences between measured product concentrations $y_i(t_j)$ and respective model predictions $z_i(t_j, k_i)$ for each substrate $i = 1, ..., 256$ and all time points $t_j$, $j = 1, ..., n$. The Matlab algorithm fmincon was used to solve this optimization problem with initial values $k_i = 1$ for all $i$ and $t_j = 0.5$, $j = 2, ..., n$ and a maximum of function evaluations set to 100,000 to ensure convergence.

**HPLC-MS/MS analysis of the oxidation products**. To determine the activity of TET1 and TET2, in vitro oxidation reactions were performed using double-stranded 30mer oligonucleotides containing a single hemimethylated or hemi-hydroxymethylated CpG site in different flanking contexts (Supplementary Table 4). Oxidation reactions were carried out at 37 °C for varying time intervals using 0.5 µM DNA in the same buffer as in the Deep Enzymology reactions and a total reaction volume of 20 µL. Reactions were stopped by freezing in liquid nitrogen and the enzymes were inactivated with Proteinase K for 1 h at 50 °C. Water was added to obtain a final volume of 35 µL and the samples were prepared for HPLC-MS/MS analysis via enzymatic digestion and filtration as described in ref. [51]. Enzyme concentrations of TET1 and TET2 were 1.6/2.0 µM for the favored substrates and 3.2/8 µM for the disfavored substrates.

The following external standards were used: unmodified dATP, dGTP, dCTP, and dTTP were from Genaxxon bioscience GmbH (Ulm, Germany), 5fdCTP, 5cadCTP, and 5mdCTP were from tebu-bio GmbH (Offenbach, Germany), and 5hmdCTP was from Zymo Research Europe GmbH (Freiburg, Germany). Samples and external standard mixtures at appropriate concentrations were diluted 1:1 with 1% acetonitrile, 0.1% formic acid prior to analysis. Data were acquired on an Impact II quadrupol time-of-flight mass spectrometer (Bruker Daltonic, Bremen, Germany) using an ESI source, coupled to an Infinity II HPLC system (Agilent Technologies, Waldbronn, Germany). Separation of analytes by liquid chromatography was carried out on a Purospher® STAR RP-8 endcapped Hibar® RT column (3 × 150 mm, 3 µm, Merck KGaA, Darmstadt, Germany). At a full duty cycle time of 18 min a gradient of water and acetonitrile, including 0.1% formic acid, was run at a flow rate of 250 µL/min at a constant temperature of 45 °C. The initial concentration of 1% acetonitrile was maintained for about 5 min, prior to raising the organic content up to 100% by a linear gradient within 5 min. This condition was maintained for 3 min before returning to 1% acetonitrile within 0.1 min for equilibration of the column, lasting for 4.9 min. The injection volume was 60 µL for the analyzed samples and external standard mixtures. Samples were stored in the autosampler at 10 °C.

Mass spectrometry data were acquired under the control of Bruker otofControl (version 4.1) and Bruker Compass HyStar (version 4.1). The mass range was set from 50 to 450 m/z and the spectra rate was set to 2 Hz. The source parameters were adjusted to the flow rate: endplate offset, 700 V; capillary, 4500 V; nebulizer gas, 2.5 bar; dry gas flow rate, 6.0 L/min, and dry gas temperature, 200 °C. Ion transfer settings were ion energy, 4 eV; collision energy, 5 eV; collision RF, 650 Vpp; transfer time, 80 µs and prepulse storage, 5 µs. Data were acquired in positive pseudo MRM mode by isolation of specified m/z ratios of the precursor ions and recording of product ion spectra. The isolation mass range width for fragmentation experiments was set to 1.0 m/z and the collision energy was optimized for the analytes as follows: 15 eV for dC, 5mdC and 5hmdC, and 20 eV for dA, dG, dT, 5fdC and 5cadC. Data analysis was carried out under the assistance of Skyline (version 20.2) by calculation of the peak area of the transitions given in Supplementary Fig. 6. Depending on the injection volume the integrals were normalized for all reactions and plotted against the reaction time using Excel. For further normalization, the integrals were first divided by the area obtained for cytosine nucleosides in each reaction, which do not participate in the oxidation pathway. Afterward, calibration factors obtained from the external calibration curves were applied and the sum of all modified cytosine nucleosides was set to 1 for each reaction.

**Fitting of reaction progress curves by numerical integration**. Reaction progress curves for the oxidation of 5mC and 5hmC containing substrates were analyzed by fitting to theoretical reaction progress curves calculated by numerical integration using rate constants $k_1$ for the conversion of 5mC to 5hmC, $k_2$ for the conversion of 5hmC to 5fC, $k_{12}$ for the direct, processive conversion of 5mC to 5fC, and $k_3$ for the conversion of 5fC to 5caC. The following set of differential equations (Eqs. 3–6) was used:

$$d(c_{5mC})/dt = -k_1 c_{5mC} - k_{12} c_{5mC} \qquad (3)$$

$$d(c_{5hmC})/dt = k_1 c_{5mC} - k_2 c_{5hmC} \qquad (4)$$

$$d(c_{5fC})/dt = k_{12} c_{5mC} + k_2 c_{5hmC} - k_3 c_{5fC} \qquad (5)$$

$$d(c_{5caC})/dt = k_3 c_{5fC} \qquad (6)$$

Numerical integration was conducted in Excel with a time step of 0.01 min. Baseline and individual intensity factors were included. Fitting of the data was performed using the solver module by minimizing the RMSD between theoretical and experimental data points.

**Analysis of genomic 5mC and 5hmC data**. Global DNA methylation and hydroxymethylation patterns in human lung and liver cells were taken from published whole-genome bisulfite data (GEO accession number GSE70091, data sets N1 and N2 for both lung and liver tissue)[10]. CpG sites of the + strand were filtered for coverage ≥10 using a local instance of the Galaxy server. To retrieve the DNA sequences flanking the CpG sites BEDTools GetFastaBed[52] was used with the hg19 reference sequence. The same analysis steps were performed for data sets obtained after bisulfite or oxidative bisulfite sequencing of individual tissue samples and after joining the difference was calculated to obtain the methylome as well as the hydroxymethylome of both cell types. Afterward, average 5hmC levels in all NNCGNN flanks were determined with a home written program. Excel was used to obtain the Pearson correlation factors. The N1 and N2 data sets were highly correlated and merged for further analysis.

Changes of DNA methylation after knock-out of TET1 or TET2 in mouse embryonic stem cells were taken from published reduced representation genome bisulfite data (GEO accession number GSE122814)[34]. Data of different repeats were combined and CpG sites filtered that are covered in wt ES cells and TET1 KO (345590 sites) or wt ES cells and TET2 KO (216664 sites). For these sites, the change in methylation was calculated $\Delta_m = X_m(KO) - X_m(wt)$. Overall, average $\Delta_m$ values were 0.81% for TET1 KO and 1.7% for TET2 KO. The relative enrichment and depletion of bases was determined in Top 10,000 sites showing a gain of methylation. Weblogos were prepared using WebLogo 3 (http://weblogo.threeplusone.com/)[53]. The correlation of local 5hmC levels with average TET preferences were determined using GSE70091 data sets N1. 5hmC levels calculated to be <0 in the oxidative bisulfite data were set to 0. Pearson R-values were determined for sliding regions of 18 consecutive CpG sites in arbitrarily chosen parts of chromosomes 2, 5, 9, 10, and 17.

**EMSA with UHRF2 as 5hmC reader**. A histidine-tagged expression construct of the hUHRF2 SRA domain (amino acid 419–648) was obtained from Dr. Rui-Ming Xu (Chinese Academy of Sciences, Beijing, China)[15]. The protein was over-expressed using BL21 (DE3) Codon+ RIL E. coli cells (Stratagene). The cells were grown in LB medium until an $A^{600,nm}$ of 0.5 was reached and overexpression was induced for 12–14 h at 16 °C by the addition of 0.5 mM isopropyl-1-thio-D-galactopyranoside. Ni-NTA affinity chromatography was used for purification as described for the TET enzymes, but with adjusted sonication/wash buffer (20 mM Tris-HCl pH 7.2, 500 mM NaCl, 0.2 mM DTT, 30 mM imidazole, and 10% glycerol) and elution buffer (20 mM Tris-HCl pH 7.2, 500 mM NaCl, 0.2 mM DTT, 500 mM imidazole, and 10% glycerol). Aliquots of the protein were stored at

−80 °C in dialysis I buffer (20 mM Tris-HCl pH 7.2, 200 mM KCl, 0.2 mM DTT, and 10% glycerol). The quality of the protein was verified via Coomassie-stained SDS gels and the concentration was determined using Nano-Drop (Thermo Scientific). For the DNA binding assays, the different amounts of the purified hURF2 SRA protein were incubated with 0.5 μM the 30 bp long TET1 substrates containing an unmodified, hemimethylated or hemihydroxymethylated CpG site (Supplementary Table 4) in binding buffer (20 mM HEPES pH 7.9, 100 mM NaCl, 2 mM $MgCl_2$, 2 mM DTT, 0.1% NP-40, and 5% glycerol) in a total volume of 10 μL. After 30 min incubation on ice, 5 μL of the samples were directly loaded on a 6% Polyacrylamide gel with 0.25X TBE Buffer (22 mM Tris-HCl pH 8.0, 22 mM boric acid, 0.5 mM EDTA), which was pre-run for 40 min at 8 °C with 200 V. Electrophoresis was performed for another 40 min at 100 V and the gel was stained with GelRed (Genaxxon).

**Statistics and reproducibility**. The number of independent experimental repeats is indicated for each experiment. Pearson correlation factors were determined with MS Excel. Additional statistical tests were not applied.

**Reporting Summary**. Further information on research design is available in the Nature Research Reporting Summary linked to this article.

## Data availability

Global DNA methylation and hydroxymethylation patterns in human lung and liver cells were taken from published whole-genome bisulfite data (GEO accession number GSE70091, data sets N1 and N2 for both lung and liver tissue)[10]. DNA methylation after knock-out of TET1 or TET2 in mouse embryonic stem cells were taken from published reduced representation genome bisulfite data (GEO accession number GSE122814)[34]. NGS kinetic raw data generated in this study (including raw Fastq files and extracted sequences) are available at DaRUS under https://doi.org/10.18419/darus-2114. Source data and uncropped images are provided in Supplementary Data 2 and Supplementary Fig. 11. All other data are available from the corresponding author upon reasonable request.

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

## Acknowledgements
This work has been supported by the Deutsche Forschungsgemeinschaft grant (JE252/36 to AJ, RA1840/2-1 to NR, and under Germany's Excellence Strategy – EXC 2075 390740016).

## Author contributions
SA conducted the biochemical work. J Bräcker, BO, and J Brockmeyer conducted the LC-MS experiments. VK and NR conducted the monoexponential fitting of NGS reaction progress curves. SA, PB, and AJ conducted the bioinformatic work and fitting of the LC-MS data. AJ, PB, J Brockmeyer, and NR supervised the work. All authors were involved in data analysis and interpretation. AJ and SA prepared the manuscript with input from all authors. All authors approved the final version of the manuscript

## Funding

## Competing interests
The authors declare no competing interests.
