## [Peer Review File · Communications Biology]

Reviewers' comments:

Reviewer #1 (Remarks to the Author):

This manuscript reported that the flanking sequences could influence the activity of TET1 and TET2 methylcytosine dioxygenases and affect genomic 5hmC patterns. The data verified that the pronounced effect of the flanking sequences from -3 to +2 position on the catalytic activity of both TET enzymes, with preference for A and disfavor for G at the -1 site and disfavor for C at the +1 site. The flanking sequence preferences were also similar for 5mC and 5hmC containing CpG and non-CpG substrates. The results indicated that flanking sequences preferences could represent an important parameter that influenced genomic DNA modification patterns. In general, the manuscript is well organized and the data support the conclusions. I recommend for publication after addressing the following issues.

1. The author used a deep enzymology approach to investigate the flanking sequence effects on TET activities, how to confirm that the DNA binding pocket of TET was about 9 base pairs?
2. Deep enzymology workflow has been utilized to study the substrate specificity of DNA MTases in random sequence context (Nature Communications, 2020, 11, 3355). What is the innovation of this method in studying the flanking sequence effects on TET activities?
3. The author used LC-MS to confirm the flanking sequence preferences on the effects of TET activities, could the oxidation levels be validated by other quantitative methods, such as WB experiments that could identify 5hmC or other higher oxidized forms?

Reviewer #2 (Remarks to the Author):

The Jeltsch lab has developed a technique called "Deep Enzymology" whereby different oligonucleotide sequences are incubated with a DNA modifying enzyme of interest, and subsequently the modified DNA is assessed by high throughput sequencing. This is on the one hand very powerful, as this technique allows for "one shot" assessment of enzyme preference for thousands of different nucleotide sequences, which is orders of magnitude more efficient than traditional biochemical approaches. On the other hand there are obvious drawbacks, as the technique neglects protein partners and chromatin context (eg, nucleosome positioning and histone modifications) that may impact enzyme activity. Nevertheless, the Jeltsch lab has already successfully implemented this approach to study DNA methyltransferases. It was natural that they extended Deep Enzymology to the study of TET enzymes, which oxidize 5mC to hydroxymethyl-, formyl- and finally carboxylcytosine. 5fC and 5caC are converted to uracils after bisulfite conversion, whereas 5mC and 5hmC are not; thus, using pre-methylated or pre-hydroxymethylated oligos, they could incubate with the catalytic domain of TET enzymes (TET1 and TET2 here), perform bisulfite conversion, and assess efficiency. Their findings were bolstered by a LC-MS approach. Most importantly, comparison of their Deep Enzymology data with public 5hmC data strongly confirms the validity of the technique.

Overall, I found this to be a very solid study. I would say that the results only incrementally advance our knowledge of TET activity, as several genome-wide data sets already can provide insight into sequence preference. That said, given that TET1 contains a CxxC domain (which binds to CpG rich regions), and TET2 has been implicated in a number of studies as interacting with transcription factors, it is not necessarily intuitive that the catalytic domain itself can confer specificity—however mild. Therefore, it is useful information for the TET and DNA methylation fields. Understanding the biological importance of these sequences (both the ones amenable and those more resistant to TET modification) will be a worthwhile avenue of research.

NB: I am not a biochemist by training, so I will let other reviewers assess the biochemical assays performed and the subsequent analysis.

Major point

- Something that wasn't discussed to me: what was the rationale for using hemimethylated

substrates, when so much of genomic methylation is symmetric. Can the authors elaborate for why they made this choice: was it simply for technical complications during second strand synthesis? I think this is potentially a fair rationale. That said, they could test if symmetric methylation increases TET activity in the sequences most enriched for oxidation after Deep Enzymology.

Minor points

- The authors describe how starting with 5mC and 5hmC substrates will help understand the transition from 5mC > 5hmC, however they should clarify in the text that this will not help understand fC and caC transition, as this is nebulous the way the sentence was written. Correct me if I'm wrong.

- Fig S4 shows the different reaction rates inferred from sequencing data, but the sequence with fastest "rate" isn't the same as the top hits for "activity" in figure 2. Could the authors explain.

- I think a supp figure with Tet CD constructs used will be more clarifying than the description in the text

Reviewer #3 (Remarks to the Author):

In this manuscript, Albert Jeltsch and his colleagues used a so-called "deep enzymology" method to study the effect of neighboring DNA sequence variation on the in vitro activities of Tet1 and Tet2. They investigated random variation of 256 NNCGNN sequences on both 5mC and 5hmC. They validated their observation with in vitro enzymatic assays on two substrates (one favored and the other disfavored), and analyzed published genomic data as well as structural information. Their study is quite complete.

The authors now have analyzed flanking sequences of enzymes working on DNA CpG methylation and 5-hydroxymethylation, including previously published de novo activity of Dnmt3A/3B, maintenance activity of DNMT1, and now in the current manuscript, demethylase activity. Do these enzymes have the same preferred flanking sequences? If this is not the case, the differences in the preferred flanking sequences among the three enzyme activities are worthy of a thoughtful discussion. What could be the biological significance for the three enzyme activities having similar and/or different flanking sequence preferences, when the substrates of maintenance activity are the product of de novo activity and the substrates of the demethylases are the product of DNA methyltransferases?

Reply to the reviewer's comments

Reviewer #1 (Remarks to the Author)

“This manuscript reported that the flanking sequences could influence the activity of TET1 and TET2 methylcytosine dioxygenases and affect genomic 5hmC patterns. The data verified that the pronounced effect of the flanking sequences from -3 to +2 position on the catalytic activity of both TET enzymes, with preference for A and disfavor for G at the -1 site and disfavor for C at the +1 site. The flanking sequence preferences were also similar for 5mC and 5hmC containing CpG and non-CpG substrates. The results indicated that flanking sequences preferences could represent an important parameter that influenced genomic DNA modification patterns. In general, the manuscript is well organized and the data support the conclusions. I recommend for publication after addressing the following issues.”

Reply: Thank you very much for this positive assessment.

“1. The author used a deep enzymology approach to investigate the flanking sequence effects on TET activities, how to confirm that the DNA binding pocket of TET was about 9 base pairs?”

Reply: Thank you for pointing to this ambiguity. The sentence has been rewritten and the claim of 9 base pairs removed.

“2. Deep enzymology workflow has been utilized to study the substrate specificity of DNA MTases in random sequence context (Nature Communications, 2020, 11, 3355). What is the innovation of this method in studying the flanking sequence effects on TET activities?”

Reply: In this paper, we apply our newly developed Deep Enzymology approach for the first time on TET enzymes, thereby providing the first systematic data describing the flanking sequence preferences of TET enzymes. Hence, the novelty of our work is not based on methodological developments but on our new findings. Previously, the commonly accepted state of knowledge was that TET enzymes do not have flanking sequence preferences {Hu, 2015; Yin, 2016}. However, this conjecture was based on the investigation of catalytic activities using only a small number of different substrates. We have added one sentence to the discussion to make this point clear.

“3. The author used LC-MS to confirm the flanking sequence preferences on the effects of TET activities, could the oxidation levels be validated by other quantitative methods, such as WB experiments that could identify 5hmC or other higher oxidized forms?”

Reply: Please note that in our manuscript, mass spectrometry was used to validate the Bisulfite conversion – NGS data, so mass spec was already the second technique, not the primary one. In principle, the bisulfite conversion – NGS data also “validate” the mass spectrometry data. We like to mention that mass spectrometry is widely accepted as a quantitative technology to detect DNA base modifications and it has the additional advantage that it detects not only 5hmC but also the higher oxidized forms of 5hmC.

We assume that the hint towards WB refers to antibody-based methods of mC, hmC, fC and caC detection. We have applied these in the initial phase of the project and found them not quantitative and definitely unsuitable to detect 2- or 3-fold differences in catalytic rates. Moreover, they will not allow to monitor all 4 relevant species together, but only in separate blots. A survey of newer

literature confirms that quantitative studies with TET enzymes usually are based on mass spectrometry, not on antibody-based detection methods.

We have added one sentence to the discussion to clarify this point.

Reviewer #2 (Remarks to the Author)

“The Jeltsch lab has developed a technique called “Deep Enzymology” whereby different oligonucleotide sequences are incubated with a DNA modifying enzyme of interest, and subsequently the modified DNA is assessed by high throughput sequencing. This is on the one hand very powerful, as this technique allows for “one shot” assessment of enzyme preference for thousands of different nucleotide sequences, which is orders of magnitude more efficient than traditional biochemical approaches. On the other hand there are obvious drawbacks, as the technique neglects protein partners and chromatin context (eg, nucleosome positioning and histone modifications) that may impact enzyme activity.”

Reply: We fully agree with the reviewer that the activity of TET enzymes at a defined genomic locus depends on their locus specific targeting and their regulation by chromatin modifications at the target regions, post-translational modifications or interacting proteins. The novelty of our approach lies in the discovery that the catalytic domain of the TET enzymes in addition has strong flanking sequence preferences. Hence, after recruiting the enzyme to a target locus, activating it by PTMs and binding of complex partners, the local CpG-site specific activity depends on the flanking sequence preferences. This is clearly indicated in our work by the strong correlation of TET enzyme flanking sequence preferences and genome wide NN^{5hm}CGNN patterns as well as DNA demethylation rates in NN^{5m}CGNN contexts. We have rewritten parts of the discussion to make this point more obvious and explain our arguments more accurately.

“Nevertheless, the Jeltsch lab has already successfully implemented this approach to study DNA methyltransferases. It was natural that they extended Deep Enzymology to the study of TET enzymes, which oxidize 5meC to hydroxymethyl-, formyl- and finally carboxylcytosine. 5fC and 5caC are converted to uracils after bisulfite conversion, whereas 5meC and 5hmC are not; thus, using pre-methylated or pre-hydroxymethylated oligos, they could incubate with the catalytic domain of TET enzymes (TET1 and TET2 here), perform bisulfite conversion, and assess efficiency. Their findings were bolstered by a LC-MS approach. Most importantly, comparison of their Deep Enzymology data with public 5hmC data strongly confirms the validity of the technique. Overall, I found this to be a very solid study.”

Reply: Thank you very much for this positive assessment.

“I would say that the results only incrementally advance our knowledge of TET activity, as several genome-wide data sets already can provide insight into sequence preference. That said, given that TET1 contains a CxxC domain (which binds to CpG rich regions), and TET2 has been implicated in a number of studies as interacting with transcription factors, it is not necessarily intuitive that the catalytic domain itself can confer specificity—however mild. Therefore, it is useful information for the TET and DNA methylation fields. Understanding the biological importance of these sequences (both the ones amenable and those more resistant to TET modification) will be a worthwhile avenue of research.”

Reply: The novelty of our work is in the systematic analysis of the flanking sequence preferences of the catalytic parts of TET1 and TET2, which has not been conducted before. The very clear effects of these preferences on genome wide NN^{5hm}CGNN patterns and DNA demethylation in NN^{5m}CGNN contexts shown in our paper identify these factors as novel parameters that influence genome wide DNA methylation and DNA hydroxymethylation patterns. We like to mention that the strong flanking sequence preferences of TET1 and TET2 are described here for the first time. Previously, the commonly accepted state of knowledge was that TET enzymes do not have flanking sequence preferences {Hu, 2015; Yin, 2016}. However, this conjecture was based on the investigation of catalytic activities on a small number of different substrates. We have added on sentence to the discussion to make this point clear.

“Major point

- Something that wasn't discussed to me: what was the rationale for using hemimethylated substrates, when so much of genomic methylation is symmetric. Can the authors elaborate for why they made this choice: was it simply for technical complications during second strand synthesis? I think this is potentially a fair rationale. That said, they could test if symmetric methylation increases TET activity in the sequences most enriched for oxidation after Deep Enzymology.”

Reply: Thank you for this interesting comment. In fact it would be possible to generate fully methylated DNA by enzymatic methylation of the randomized substrate with M.SssI followed by another purification step. However, using fully methylated DNA would create a very complex situation, because the TET enzymes would then act on both DNA strands. This means that oxidation would occur in complex mixtures of substrates with the second strand either carrying 5mC, 5hmC or even higher oxidized cytosine bases. Measuring the flanking sequence dependent oxidation rates of a 5mC in the target strand and a 5mC, 5hmC, 5fC and 5caC context on the non-target strand would require a much higher sequencing depth. This problem is amplified by the fact that for random sites typically activity is more preferred on one strand than on the other due to the non-palindromic nature of the flanking sequence preferences. More precisely, for oxidation of good sites activity will typically be favored on the target strand, but this is not the case for disfavored flanks, where activity will be preferred on the non-target strand in most cases. Therefore, oxidation events of 5mC in a good flanking context with 5hmC (or higher oxidation products) in the opposite strand will be very rare, as well as oxidation events of 5mC in an unfavorable flanking context with unoxidized 5mC in the opposite strand. We agree with the reviewer that an analysis of the influence of the modification state of the cytosine residue in the non-target strand on flanking sequence preferences will be an interesting topic for future studies. This statement has been added to the discussion of our manuscript.

“Minor points

- The authors describe how starting with 5mC and 5hmC substrates will help understand the transition from 5mC > 5hmC, however they should clarify in the text that this will not help understand fC and caC transition, as this is nebulous they way the sentence was written. Correct me if I'm wrong.”

Reply: We have rewritten the corresponding sentence on p. 3 for more clarity. In addition, one paragraph on p. 4 has also been edited for clarity. Finally, the legend in Supplementary Figure S1 has been amended to clarify that the conversion of 5hmC to 5fC is detected in our assays.

“- Fig S4 shows the different reaction rates inferred from sequencing data, but the sequence with fastest “rate” isn't the same as the top hits for “activity” in figure 2. Could the authors explain. “

Reply: The reason for this design was that we tried to find “compromise” substrates that are suitable for both enzymes. Hence, the favorable substrate is the second best for TET1 and among the best 15% for TET2. The disfavored substrate is the worst for TET1 and among the worst 5% for TET2. This point had been mentioned on p. 5 of our original manuscript “...which contain 5mC or 5hmC in a favorable (TACGTA, rank 2 of 256 for TET1 and 29 of 256 for TET2, where low numbers indicate high preference) or unfavorable sequence context (CGCGCC, rank 256 for TET1, rank 244 for TET2).

“- I think a supp figure with Tet CD constructs used will be more clarifying than the description in the text”

Reply: Thank you. This has been provided as proposed now as Supplementary Figure S10.

Reviewer #3 (Remarks to the Author)

“In this manuscript, Albert Jeltsch and his colleagues used a so-called “deep enzymology” method to study the effect of neighboring DNA sequence variation on the in vitro activities of Tet1 and Tet2. They investigated random variation of 256 NNCGNN sequences on both 5mC and 5hmC. They validated their observation with in vitro enzymatic assays on two substrates (one favored and the other disfavored), and analyzed published genomic data as well as structural information. Their study is quite complete.”

Reply: Thank you for this positive assessment.

“The authors now have analyzed flanking sequences of enzymes working on DNA CpG methylation and 5-hydroxymethylation, including previously published de novo activity of Dnmt3A/3B, maintenance activity of DNMT1, and now in the current manuscript, demethylase activity. Do these enzymes have the same preferred flanking sequences? If this is not the case, the differences in the preferred flanking sequences among the three enzyme activities are worthy of a thoughtful discussion. What could be the biological significance for the three enzyme activities having similar and/or different flanking sequence preferences, when the substrates of maintenance activity are the product of de novo activity and the substrates of the demethylases are the product of DNA methyltransferases?”

Reply: Thank you for this interesting comment. We agree with your conclusion, that the flanking sequence preferences of TET and DNMTs together influence methylation patterns. This was already stated in the last sentence of the discussion of our manuscript: “Future work needs to address the details of the crosstalk of all these processes which together determine global DNA modification patterns and epigenetic information transfer.” Moreover, we have shown that the prediction of cellular 5hmC profiles was improved by the combination of data on 5mC density and TET flanking sequence preferences (Fig. 5C and D).

Based on your comments we now also prepared a comparison of the flanking sequence preferences of TET1, TET2, DNMT1, DNMT3A and DNMT3B which has been included as Supplementary Figure S5 and described in the results section on p. 4 and 5. This analysis shows that the flanking sequence preferences of TET1 and TET2 are highly similar. Strikingly, the preferences of TET, DNMT3A, DNMT3B and DNMT1 are very different, which is not surprising as these enzymes use different mechanisms for the DNA recognition.

REVIEWERS' COMMENTS:

Reviewer #1 (Remarks to the Author):

The authors have made corresponding changes according to the comments. The concerns have been well addressed. I am now satisfied with the manuscript and would like to recommend publication in Communications Biology.

Reviewer #2 (Remarks to the Author):

The authors satisfactorily responded to my comments. I endorse for publication.

Reviewer #3 (Remarks to the Author):

I have no further questions/comments.